# Variability in Gaseous Elemental Mercury at Villum Research Station, Station Nord, in North Greenland from 1999 to 2017

[1]*Henrik Skov, [1]Jens Hjorth, [1]Claus Nordstrøm, [1]Bjarne Jensen, [1]Christel Christoffersen, [1]Maria Bech Poulsen, [1,2]Jesper Baldtzer Liisberg, [3]David Beddows, [4]Manuel Dall'Osto and [1]Jesper Christensen

*Corresponding author, [1]Department of Environmental Science, iClimate, Aarhus University, Frederiksborgvej 399, 4000 Roskilde, Denmark. [2]Physics of Ice, Climate and Earth, University of Copenhagen, Tagensvej 16, 2200 København N, Denmark. [3]Centre for Atmospheric Science, Division of Environmental Health & Risk Management, School of Geography, Earth & Environmental Sciences, University of Birmingham, Edgbaston Birmingham, B15 2TT, United Kingdom. [4]Institute of Marine Sciences (ICM), Consejo Superior de Investigaciones Científicas (CSIC), Pg. Marítim de la Barceloneta 37–49, 08003, Barcelona, Spain.

*Correspondence to*: Henrik Skov (HSK@ENVS.AU.DK)

**Abstract.** Mercury is ubiquitous in the atmosphere and atmospheric transport is an important source for this element in the Arctic. Measurements of gaseous elemental mercury (GEM) have been carried out at Villum Research Station (Villum) at Station Nord, situated in north Greenland. The measurements cover the period 1999–2017 with a gap in the data for the period 2003–2008 (for a total of 11 years). The measurements were compared with model results from the Danish Eulerian Hemispheric Model (DEHM) model that describes the contribution from direct anthropogenic transport, marine emission and general background concentration. The percentage of time spent over different surfaces was calculated by back-trajectory analysis and the reaction kinetics were determined by comparison with ozone.

The GEM measurements were analysed for trends, both seasonally and annually. The only significant trends found were negative ones for the winter and autumn months. Comparison of the measurements to simulations using the Danish Eulerian Hemispheric Model (DEHM) indicated that direct transport of anthropogenic emissions of mercury accounts for between 14 and 17% of the measured mercury. Analysis of the kinetics of the observed Atmospheric Mercury Depletion Events (AMDEs) confirms the results of a previous study at Villum of the competing reactions of GEM and ozone with Br, which suggests a lifetime of GEM of about a month. However, a GEM lifetime of 12 months gave the best agreement between model and measurements. The chemical lifetime is shorter and thus the apparent lifetime appears to be the result of deposition followed by reduction and reemission; for this reason the term 'relaxation time' is preferred to 'lifetime' for GEM. The relaxation time for GEM causes a delay between emission reductions and the effect on actual concentrations.

No significant annual trend was found for the measured concentrations of GEM over the measurement period despite emission reductions. This is interesting, and together with low direct transport of GEM to Villum as found by the DEHM model, it shows that the dynamics of GEM are very complex. Therefore, in the coming years intensive measurement networks are much needed to describe the global distribution of mercury in the environment as the use of models to predict future levels will still be highly uncertain. The situation is increasingly complex due to global change that most likely will change the transport patterns of mercury not only in the atmosphere but also between matrixes.

# 1 Introduction

The effects of long-range atmospheric transport of anthropogenic pollutants into the Arctic are well documented: contaminants are affecting the Arctic by contamination of food chains, and by altering the radiation budget, thus contributing to climate change (UNEP 2013a; AMAP/UNEP 2013, Heidam et al. 2004, Breider et al. 2017). There are still only few local sources of pollutants in the Arctic and long-range transport mainly from mid latitudes represents the main source.

Mercury (Hg) is one of the first substances that have been identified as a pollutant in the food web worldwide, causing adverse

effects to human health and wildlife. The Minamata Convention, aiming at reducing the exposure of human beings and the environment to mercury, was signed in 2013 (UNEP 2013b) and it entered into force in 2017.

The sources of mercury in the environment can be divided into terrestrial emissions (including geogenic, biomass burning and reemissions from soils and vegetation), anthropogenic and oceanic emissions accounting for 2.1, 2.5 and 3.4 ktonnes of the emissions, respectively (Outridge et al. 2018). This is in good agreement with other estimates. The global anthropogenic

emissions of mercury were estimated as 2.5 ktonnes in 2010 (UNEP 2013a; AMAP/UNEP 2013) and including the large uncertainty on these numbers, they are not significantly different. According to an estimate by (Pirrone et al. 2010) natural sources and reemission processes (hereafter referred to as ´background sources´), accounted for 5207 Mg per year in 2005 while the amount of new anthropogenic inputs is 2320 Mg per year, also close to the latest emission estimate (Outridge et al. 2018). According to recent assessments (Pacyna et al. 2010; Pirrone et al. 2010; AMAP/UNEP 2013; UNEP 2013; Muntean

et al. 2014), the main anthropogenic sources of atmospheric mercury are coal combustion and artisanal/small gold mining, with relevant contributions from non-ferrous metal smelting and iron and steel production along with several other industrial/residential sources such as waste incineration. The main background source is evasion from ocean surfaces, accounting for about half of the sum of the natural and reemission contributions (Pirrone et al. 2010). Reemission of deposited atmospheric mercury of anthropogenic origin gives a major contribution to the reemission budget, e.g. it has been found that

the accumulation of mercury inputs from anthropogenic sources to oceans have led to an increase in the mercury concentration in surface waters of about a factor of three (Lamborg et al. 2014)). Mercury is transported by rivers, sea currents, and in the troposphere. Mercury in air is mainly found in the gas phase, where the major part is gaseous elemental mercury (GEM), covering more than 90%, while a minor part is oxidised mercury as well as particle bound mercury. The share of oxidised mercury of the overall global emissions of mercury has been estimated to be around 25%, based on speciation factors from the

Arctic Monitoring and Assessment Program (Muntean et al. 2018). The atmospheric lifetime of GEM has earlier been estimated to be in the range of about one year (Steffen et al. 2008), while those of oxidised forms of mercury are shorter. Theoretical and laboratory studies showed that the lifetime of GEM towards bromine-initiated oxidation is much shorter than one year (Balabanov et al. 2005; Dibble et al. 2012; Donohoue et al. 2005; Donohoue et al. 2006; Goodsite et al. 2004, 2012; Jiao and Dibble 2017). Applying the latest kinetic data, Horowitch et al. (2017) found a lifetime in the atmosphere of GEM

with respect to removal by oxidation of 2.7 months using the GEOS-CHEM model coupled to an ocean general circulation model (MITgcm). Including photoreduction, the lifetime of total gaseous mercury (TGM) was found to be 5.2 months, close

to the value 6.1 months of Holmes et al. 2010 but applying a much higher Br concentration and consequently also a faster photoreduction to reach a similar result. The deposition rate depends on the chemical processes that transform GEM into the less volatile $Hg^{II}$ species; these processes are only partially understood (Angot et al. 2016). Chemical conversion of GEM to

$Hg^{II}$ seems to be particularly important in the Arctic area, where ozone and mercury chemistry have been found to be coupled during events where both are observed at ground level to be depleted from the air. There is strong evidence that these depletion episodes are caused by the photochemical formation of bromine atoms (Skov et al. 2004; Goodsite, Plane, and Skov 2004, 2012a; Skov et al. 2006; Kamp et al. 2018) and recently direct evidence was found for bromine-initiated AMDEs and ODEs (Wang et al. 2019).

The geographical distribution of the emissions has changed in the last decades, where Asian countries have gained importance compared to emissions in Europe, North America, and Japan. Today China accounts for about 40% of the global Hg emission (Muntean et al. 2014; Streets et al. 2019, 2017, 2018). In North America, Europe and on the North Atlantic Ocean there is seen a decline in the GEM concentration of between -1.5 and -2.2% $yr^{-1}$ (Zhang et al. 2016). In the Arctic, the decline is zero at Svalbard (Berg et al. 2013) and -0.9% at Alert (Cole et al. 2013).

The aim of the present article is to present and discuss the long time series of GEM measurements at Villum Research Station at Station Nord in North Greenland with a focus on observed inter-annual and seasonal trends as well as the likely explanations for these in terms of sources, transport patterns and dynamics.

## 2 Experimental section

### 2.1 Measurements

Villum Research Station (Villum) at Station Nord in North Greenland is the second most northerly, permanently open station in the Arctic, only preceded by Alert, Canada. The station has all the logistic requirements and infrastructures that are necessary for being a major international platform for scientific studies focused on the Arctic cryosphere, nature and interaction with humans. It is located in the farthest north-eastern corner of Greenland on the north-south oriented peninsula Princess Ingeborgs Halvø (81°36' N 16°40' W) whose northern end is a 20 x 15 $km^2$ Arctic lowland plain (see Figure 1). Villum is an important

logistic site for many scientific research activities in the Greenlandic National Park, in North Greenland, see www.villumresearchstation.dk. Ozone and GEM were measured at Flygers Hut from 1996 and 1999, respectively, until 2014 when the measurements were moved to the newly built Air Observatory (Figure 2), and they continue to this day.

Since 1999, GEM has been measured by a TEKRAN 2537 mercury analyser. In the first years, funding was only available for six months per year of observations and thus the data coverage over the entire year is limited to spring, summer and early

autumn except for the very first year. There are no measurements available for the years 2003–2008 as the research station was closed. Several generations of the instrument have been used (A, B and X versions) but we estimate that the uncertainty of measuring GEM has remained unchanged during the years as they are all calibrated towards the same standard based on the vapour pressure of $Hg^0$ using Tekran 2505 calibration unit. The principle of the instruments is as follows: a measured volume

of sample air is drawn through a gold trap that quantitatively retains elemental mercury. The collected mercury is desorbed thermally from the gold trap and is transferred by argon into the detection chamber, where the amount of mercury is detected by cold vapour atomic fluorescence spectroscopy. The detection limit is 0.1 ng m$^{-3}$ and the reproducibility for concentrations above 0.5 ng m$^{-3}$ is within 20 % based on parallel measurements with two TEKRAN 2537A mercury analysers (at a 95 % confidence interval) using the principle described in ISO Guide 98-3:2008 Uncertainty of measurement — Part 3: Guide to the expression of uncertainty in measurement (GUM:1995). The calibration of the instrument is checked every 25 hours by adding known quantities of elemental mercury to the detection system from an internal permeation source. The sample air was either taken through a sample tube heated to 50°C or by drawing sample air from a 20 cm i.d. stainless sample tube. The flow rate in the stainless tube was > 1 m$^3$ min$^{-1}$. Comparison of measurements from the two different sample lines did not reveal any difference within the uncertainty of the instruments. Prior to entering the instrument, air passes a soda lime trap to avoid passivation of the gold traps.

Ozone has been measured since 1996. Though different instruments have been applied, the measurement uncertainty is unchanged as the basic principle in all instruments is absorption of UV light at 254 nm. The stability of the instruments is ensured by addition of known concentrations of ozone from an internal ozone generator traceable to a primary standard. The uncertainty at a 95 % confidence level is <7% for concentrations above 20 ppbv and 1.4 ppbv for concentrations below 20 ppbv.

The calculation of inter-annual trends was performed applying the non-parametric Mann-Kendahl test and Sens slope calculation, using the program developed by Salmi et al. (2002).

## 2.2 Model calculations

We have applied the Danish Eulerian Hemispheric Model (DEHM) to calculate the concentrations and direct contributions from different source areas to the concentration levels in air at Villum as a function of a prescribed chemical lifetime of Hg$^0$ and the meteorological variability of the atmospheric transport from source areas.

DEHM is a three-dimensional, offline, large-scale, Eulerian, atmospheric chemistry transport model (CTM) developed to study long-range transport of air pollution in the Northern Hemisphere with focus on the Arctic or Europe. The model domain used in previous studies covers most of the Northern Hemisphere, discretised on a polar stereographic projection, and includes a two-way nesting procedure with several nests with higher resolution over Europe, Northern Europe and Denmark or the Arctic (Frohn, Christensen, and Brandt 2002; Brandt et al. 2012).

DEHM was originally developed in the early 1990's to study the atmospheric transport of sulphur and sulphate into the Arctic (Christensen 1997; Heidam, Wåhlin, and Christensen 1999; Heidam et al. 2004) and has been used to study transport of mercury to the Arctic (Christensen et al. 2004, Skov et al. 2004).

The model system has been setup with one model domain with 150x150 grid points. The domain covers the Northern Hemisphere with a grid resolution on 150 km at 60°N. The vertical grid is defined using the σ-coordinate system, with 29 vertical layers extending up to a height of 100 hPa.

The DEHM model is driven by meteorological data from the Advanced Research WRF version 3.6 (WRF ARW) (Skamarock et al., 2008). This WRF model simulation was driven by global meteorological ERA-Interim data, which is a global atmospheric reanalysis data set from the European Centre for Medium-Range Weather Forecasts (ECMWF) starting from 1979 and continuously updated in real time. These data have been inserted every 6 hours into the WRF model. The WRF model has been run in a climate mode setup, e.g. continuously updating sea surface temperature and deep soil temperature (both from the ERA interim).

The global historical AMAP Hg emissions inventories for 1990-2010 have been used as the anthropogenic emissions (UNEP 2013) for the model run with variable emissions. The 1990 emissions have been used for the model calculations for the period 1990–1992, 1995 emissions for the years 1993–1997, 2000 emissions for 1998–2002, 2005 emissions for 2003–2007 and finally the 2010 emissions for 2008–2017. The emissions for 2005 were used for the model run with constant emissions.

Emissions of mercury from biomass burning were based on CO emissions obtained from the Global Fire Emissions Database, Version 3, (van der Werf et al. 2006; Van der Werf et al., 2003), where a fixed $Hg^0/CO$ ratio of $8\times10^{-7}$ kg $Hg^0$/kg CO was applied. Emissions from oceans are based on calculated fluxes from the GEOS-Chem model (Soerensen et al. 2010).

The system has been set up with 11 different GEM tracers, which represent eight different anthropogenic source areas (Russia, Eastern Europe, Western Europe, China, North America, Rest of Asia, Africa and South America), biomass burning, ocean sources and the prescribed boundary conditions on 1.5 ng/m$^3$ for the entire period. The latter is introduced because of the long lifetime of $Hg^0$ and accounts for the transport across the equator with the exchange velocity between the two hemispheres of about 1 year. The boundary condition concentration of 1.5 ng/m$^3$ represents the typical global background concentrations, which account for all emissions in both hemispheres, and are close to the concentrations at equator as given in Selin et al (2008). The boundary conditions were kept constant during the period covered by the model.

There have been made 2x3 different model runs covering the period from 1990 to 2017, with two main emission setups, which are with either constant anthropogenic missions (using the emissions in 2005 for all years) or the variable emissions for 1990–2010. Each emission setup is run with a simple fixed first order reaction lifetime for $Hg^0$ of 1 month, 6 months and 1 year, respectively. The model does not include Arctic mercury depletion in the runs presented here; it focuses only on the direct long-range transported mercury contribution to the GEM concentration at Villum. For each model run the contributions of the 11 different tracers are estimated in order to investigate this contribution as function of the fixed first order reaction lifetime for $Hg^0$, changing meteorology and changing emissions.

**2.3 Trajectory model**

In order to investigate the influence of different surfaces on GEM concentration, 120-hour back trajectories for air masses arriving at 100 m altitude at Villum were calculated with hourly resolution using the BADC (British Atmospheric Data Centre) Trajectory Service. For each of the trajectories, the time spent over different surfaces was calculated using a polar stereographic map of the Northern Hemisphere, where each of the 1024x1024 24 km grid cells were classified as land, sea, snow or sea ice, and thus the percentage of the total transport time spent over these four types of surfaces could be calculated. The snow and

ice coverage values were generated by the NOAA/NESDIS Interactive Multisensor Snow and Ice Mapping System (IMS) developed by the Interactive Processing Branch of the Satellite Services Division. For what concerns the sea ice coverage, a similar calculation was performed using daily stereographic maps of sea ice concentration with a resolution of 12.5 km, available also from NOAA/NESDIS. This calculation allowed establishing the percentage of time where the air mass of the back trajectory was passing over sea ice as done earlier in studies of atmospheric particle dynamics (Dall'Osto et al. 2018).

Combining these calculations for the periods where GEM measurements were carried out at Villum, the percentages of the 120-hour duration of the trajectory, where the air masses passed over land, sea, snow and sea ice surfaces could be established.

## 3 Results and discussion

The measurements of GEM and ozone from 1996 to 2017 are shown in Figure 3. A seasonal pattern is observed for each year, see Figure 4. In January and February, the level of ozone and GEM is rather stable. After the polar sunrise, the concentration

starts to fluctuate strongly, and ozone and GEM are depleted quickly (within 2 to 10 hours). Figure 5 shows the variations of the yearly average GEM concentration and the average for the winter season between 1999 and 2018, where only periods with more than 50% data coverage have been included. The annual averages show a negative trend, however not significant at a 90% confidence level. The autumn (September-October-November) and the winter months show both negative trends that are significant at a 90% confidence level. The trends, in percentage of the average GEM concentrations during these periods, are

-1.7%/yr for the winter period and -1.4%/yr for the autumn. The annual trend remains non-significant also when excluding the years 1999 and 2000 or the extreme value in 2017. The lack of a significant annual trend seems to be explained by the high variability of the concentrations during the spring period as well as the fact that the GEM concentration during the summer period show no evidence of a decreasing trend. This result is similar to the result obtained at Zeppelin Station on Svalbard for the period 2000 to 2008 (Berg et al. 2013) and, as previously mentioned, at Alert, Canada, where a negative trend of -0.009

ng/m$^3$ (-0.58%/yr) is seen for the period between 1995 and 2008 (Steffen et al. 2015). A study of GEM in firn snow from the Greenlandic inland ice at about 3 km altitude, Dommergue et al. (2016), showed that there is a positive trend or no trend during the period 2000–2010, though the authors pointed out that nothing can conclusively be said about the concentration trends based on their results. The behaviour of the trends may in principle be explained by changes in the emissions in the source regions, in transport patterns, in deposition, re-emission as well as atmospheric chemistry. The seasonal differences in the

trends must be explained by a different influence of these factors during the different seasons. Finally, it has been suggested that decreasing GEM concentrations in the Northern Hemisphere over the last 20 years may be partially explained by increased uptake by vegetation due to increased net primary productivity (Jiskra, 2018). Our data set does not permit to evaluate this hypothesis.

## 3.1 Changes in atmospheric chemistry

The strongest concentration trend is found during the winter, where photochemically driven chemistry obviously does not take place in the area but where long-range transport from mid-latitudes is at its maximum. The main influence of Arctic atmospheric chemistry on GEM concentrations is expected to be in the spring and summer period, where the fate of GEM is believed to depend on the presence of seasonal sea ice and the presence of air temperatures below $-4^{\circ}$ C (Christensen et al. 2004). Figure 6 shows a conceptual description of mercury removal in Arctic. A regression analysis of the number of hours

with depletion events (defined here as GEM< 0.5 ng/m$^3$) did not show any significant change over the years 2000–2017. The ozone data obtained during the period 1999–2017 also showed no significant trend for the concentrations in spring or summer. The ozone observations will be the subject of a separate publication.

The data until 2002 were used to investigate reaction kinetics of ozone and GEM with a third reactant. Log–log plots of ozone against GEM gave a straight line as seen earlier (Schroeder et al. 1998; Berg et al. 2003; Steffen et al. 2008; Skov et al. 2004).

A reaction rate for Br with Hg$^0$ was calculated, which fitted well with a reaction rate determined by theoretical chemistry (Goodsite, Plane, and Skov 2012b; Goodsite, Plane, and Skov 2004; Skov et al. 2004). We made the same analysis on the data from 2007 and onwards. GEM was averaged to a time resolution of 0.5 hours. The new analysis confirmed the previous result, though the data points were more scattered and thus the resulting slope had a higher uncertainty, mostly due to smaller difference between the initial GEM concentration and the final concentration. An important point regarding the

parameterisation of GEM depletion is that bromine-induced atmospheric mercury depletion events (AMDEs) often were observed under stagnant wind conditions and not only during situations with strong wind that may cause bromine release as proposed earlier (see latest Yang et al. 2020). Recently, the bromine-induced oxidation of Hg$^0$ has been proven directly in a study, where Br, BrO, O$_3$, GEM and RGM were measured simultaneously during AMDE and ODE and using a multiphase box model to study the complex set of processes (Wang et al., 2019).

The seasonal averaged concentration has a maximum in the summer (June-July-August) and a minimum in the spring (March-April-May). In order to test the hypothesis that the spring minimum is related to the occurrence of the combined mercury and ozone depletion events, an indicator of the duration and frequency of such depletion episodes was created. The number of measured hourly GEM concentrations below 50% of the average value in a previous event free period was compared, as a percentage, to the total number of available hourly measurements during the period of interest. For the March-April-May

period, this percentage of AMDE hours was found to be strongly correlated with the average GEM concentrations in the same period (Figure 7). Thus, there is evidence for a strong impact of AMDE on GEM concentrations in the spring period. The frequency of AMDE in spring and GEM concentration in summer showed a poor negative correlation. If the deposited Hg during AMDE should be released again during snowmelt, a positive correlation would have been expected, but this was not observed. In fact, the analyses indicate that AMDE is a net sink for mercury, which is in agreement with direct flux

measurements (Brooks et al. 2006). Interestingly, Angot et al. (2016) found a positive feedback between AMDE in spring and the concentration of GEM in summer at Alert that was attributed to reemission of mercury. Contrary to this result, even the

annual mean value at Villum had a negative correlation with AMDE hours. Though this correlation is weak, it is an indication that AMDEs affect the GEM concentration level at Villum and represent a net sink. From studies of mercury isotopes at Utqiaġvik at the North coast of Alaska (Douglas et al., 2019) and Toolik Research Station in central Alaska (Jiskra et al. 2019), it was found that most mercury in melt water was from deposition of GEM and that a large majority of deposited oxidised mercury during AMDE was reduced and reemitted. Further studies are needed to determine if these results are valid also for more northern Arctic locations as Alert, Villum or Zeppelin.

It has been determined that outflows from rivers are a main source of Hg in the Arctic Ocean (e.g. Outridge et al. 2008, Fischer et al. 2012). The present study indicates that there is an atmospheric input as well. The significance of this source depends on its chemical form. Previously atmospheric deposited mercury has been identified to be bioavailable (Moller et al. 2011) and thus might still be dominant for the mercury found in the Arctic food web.

### 3.2 Decrease in the emissions in the source regions

Recent studies show that mercury emissions from Europe and North America have been decreasing since 1990, while emissions in Asia have been increasing (UNEP 2013; Muntean et al. 2014). Russian emissions, considered as a separate entity, have been decreasing as well. Concentration data from cruises on the North Atlantic show a declining trend since 1990 with a steep decrease in the surface seawater $Hg^0$ concentration between the years 1998–2000 and 2008-2010 of -5.7% per year. It has been found that the corresponding decrease in mercury emissions from the sea can explain the decreasing trend observed over the North Atlantic and adjacent areas (Soerensen et al. 2012). Chen and co-workers (Chen et al. 2015) found that the decline in atmospheric concentrations at north-mid latitudes was significant for the period 2000–2009 but much weaker in the Arctic. They explained this by the fact that declining sea ice cover and increasing temperatures caused a tendency towards higher emissions from the sea that (partially) compensates for the forcing by decreasing surface water $Hg^0$ concentrations in the North Atlantic. The observed seasonality with significant declining tendencies in the atmospheric GEM concentration in winter (DJF) and (weaker) in autumn (SON) but not in spring (MAM) or summer (JJA) may be explained as suggested by Chen et al. The highest yearly average concentration of GEM was found in 2013, thereafter there has been a continuous decrease which may be the effect of emission reductions that now is evident also in high Arctic.

The DEHM model, using variable anthropogenic emissions as described above, shows a slightly decreasing concentration trend, -0.7% per year (see Figure 8). However, this direct anthropogenic input, assuming an atmospheric lifetime of GEM of 12 months, only accounts for between 14 and 17% of the observed GEM concentrations, (Figure 9). Including the impact of sea emissions and of the boundary conditions, and assuming a GEM atmospheric lifetime of 12 months, the model predicts an annual average GEM concentration of 1.40–1.43 $ng/m^3$, i.e. in agreement with the measured average in the period of 1.46 $ng/m^3$ , although the measured data covers a larger range of values (1.2–1.8 $ng/m^3$). When applying longer or shorter GEM lifetimes, the model results deviate more from the measured concentrations. This indicates that the best relaxation time of GEM in the Northern Hemisphere is 12 months. The chemical lifetime of GEM in the atmosphere is most likely shorter according to the theoretical and experimental evidence (e.g. Goodsite et al. 2004, 2012; Ariya et al. 2008, Donohoue et al.

2005, 2006; Dibble et al. 2012). Therefore, the deposition of Hg$^{II}$ species appears to be followed by reduction and reemission of Hg$^0$ (e.g. Brooks et al. 2006, Kamp et al. 2018, Soerensen et al. 2012, Steen et al. 2009, Cobbett et al. 2007). Thus, relaxation time seems to be a more appropriate name than lifetime for GEM. This is supported by a study on the photo-reduction of Hg$^{II}$ in cloud droplets, which was found to be much slower than the one used in models, leading to the conclusion that deposition and reemission are involved in the dynamics of atmospheric mercury (Saiz-Lopez et al. 2018). The sea emissions were found

to account for 20–21% of the GEM concentration at Villum, and the boundary conditions of 1.5 ng/m$^3$ explained 62-65%, while emissions from wildfires contributed 1% during the years of the measurements, still assuming a GEM atmospheric lifetime of 12 months.

In the calculations with DEHM, it was found that emissions from China had larger relative importance during the summer than in the winter season; however, this difference was only significant when applying relatively short (less than 1 year) atmospheric

lifetimes of GEM. The calculations for Villum were performed for the year 2001. This result agrees with Chen et al. (2018), who found that East Asia is the main source for mercury deposition in Arctic. A similar result is also reported by AMAP (AMAP 2018). Durnford et al. (2010), applying the GRAHM model, investigated the contribution of different source regions to total mercury as well as GEM concentrations at several Arctic monitoring stations at different seasons of the year. They found that for the yearly concentration averages and their variability at the Arctic stations, including Villum, Asian emissions

were the most important, accounting for more than the sum of the contributions from Europe, Russia and North America. This result is in agreement with the present study but in contrast to several studies addressing the origin of shorter-lived pollutants such as black carbon and sulphate that point to the northerly part of Eurasia as the main source regions (Nguyen et al. 2016; Freud et al. 2017). Particularly in the case of Station Nord (now named Villum Research Station), Nguyen et al. 2013 found evidence of a strong influence of direct transport of particles from Siberia including results from previous work (Heidam et al.

2004). Heidam et al. identified Russia as the main contributor to sulphate concentrations, followed by East and Western Europe, while Asian contributions appeared to be of minor importance. The explanation for this difference between modelling results regarding mercury and more short-lived air pollutants is likely to be the large difference in atmospheric lifetimes (relaxation time for GEM). The above discussion highlights the importance of assessing the chemistry of GEM and determining the fate of the resulting reaction products, especially the photo-reduction of Hg$^{II}$ compounds in marine waters.

**3.3 Changing transport patterns**

Results obtained by applying the DEHM model to simulate GEM concentrations at Villum indicate that changes in the direct atmospheric transport from source areas to Villum cannot explain the observed trend. We have found that the simulated yearly and seasonal GEM values show very little variability and no significant trend over the years 2000–2015, when the emission sources are kept constant at the 2005 level, while the meteorology is varying and treated as described above. That is opposite

to results by Dastoor et al. (2015) for a model run with constant emissions. The main reason for that is probably that processes such as chemistry and surface exchanges in Dastoor et al. (2015) are more dependent on the atmosphere and surface conditions than the simple setup in the present version of DEHM. There is better agreement between our results and Dastoor et al (2015)

for the model setup with variable emissions. We see a decrease of 0.08 ng/m$^3$ between 1992 and 2005 for Villum, while Dastoor et al. found approximately 0.1 ng/m$^3$. The study by Hirdman et al. (2010) of long term trends of sulfate and BC in the Arctic also concludes that changes in atmospheric transport only can explain a small fraction (0.3-7.2%) of the observed trends. In an earlier paper on particle formation in the Arctic atmosphere, important results have been obtained correlating the time air masses spent over different surfaces and measured concentration (Dall'Osto et al. 2018). We did the same calculations for GEM data. The correlations between the time that air masses passed over different surfaces and the measured GEM concentrations at Villum are shown in Table 1. Relatively strong negative correlations ($R^2 > 0.3$) were found only with land and sea area in the autumn. Performing a two-tailed t-test it was found that the only significant correlation at a 90% confidence was the anticorrelation in the autumn with land ($R^2 = 0.44$) while the anticorrelation with sea area was significant only at an 85% confidence level ($R^2 = 0.32$). Different types of surfaces may influence deposition and emission rates for mercury, and they may have an influence on atmospheric chemistry, e.g. by release of reactive bromine compound. However, the correlations may also not be due to a relationship caused by the impact of the surfaces within the 120 hours' time span of the trajectories but rather by the longer-term histories of the air masses. As the percentage of time passed over land by the air masses in the autumn months (SON) is very short (1-4% of the 120 hours) it seems most likely that the correlation observed is not due to a direct influence of land. It can thus be concluded from the results shown in Table 1 that no statistically significant impact of surfaces on GEM concentrations within the range of the 120 hours back trajectories could be observed.

## 4 Conclusion

In this paper, we present measurements of GEM concentrations in air at Villum Research Station from 1999 to 2017 with a break in the dataset from July 2002 until 2007. The large fraction of GEM assigned to background contribution and from sea emissions makes it difficult to assess a trend from the otherwise predicted emission reduction in the source areas for direct anthropogenic emissions of mercury. A decreasing trend in the concentration of GEM was found during autumn and winter at a 90% confidence level but it was counteracted by a weak increase during summer and a high variability during spring. Therefore, there was not any significant trend in the yearly average concentrations at the 90% confidence level.

Simulations of the concentrations at Villum using the DEHM model using a fixed emission inventory show no significant trends and thus it is concluded that the observed trends are not caused by changes in atmospheric transport patterns. The measurement area is known to be strongly influenced by long-range transport of pollutants in the winter and spring period and the only viable explanation of the observed trend in the winter appears to be decreasing emissions in the source regions. However, according to the DEHM simulations the transport of direct anthropogenic emissions only accounted for between 14 and 17 % of the GEM concentration and might be counteracted by the hemispheric background on 1.5 ng/m$^3$ that accounts for 62–65 % and was kept constant in the model. The boundary conditions represent contributions from indirect transport from sources on the Northern Hemisphere and transport from sources on the Southern Hemisphere. Sea emissions account for 20–

21%. The emissions from the North Atlantic are likely to be decreasing due to the lower mercury concentrations in the water, but a decreasing extent of the sea ice cover around Greenland may counteract this effect.

The seasonal variation confirms the effect of AMDE leading to generally lower concentrations during spring; in fact, a strong anticorrelation between the average GEM concentrations during springtime and the number of hours with AMDE conditions was observed. The analyses indicated that AMDEs are a net sink for mercury in the atmosphere and that it affects the yearly

average concentration.

Simulations with the DEHM model showed best agreement with observations applying an atmospheric lifetime of GEM of 12 months; however, it was found that the apparent lifetime is likely to be the result of a shorter chemical lifetime with respect to oxidation, followed by deposition, reduction and reemission. Thus, 'atmospheric relaxation time' seems to be a more appropriate term than 'lifetime' for GEM.

The lack of a trend in the measured concentrations of GEM despite emission reductions is striking but, together with low direct transport of GEM to Villum as found by the DEHM model, it shows that the dynamics of GEM are very complex. Therefore, in the coming years intensive measurement networks are strongly needed to describe the global distribution of mercury in the environment, because the use of models to predict future levels will still be highly uncertain. The situation is increasingly complex due to global change that most likely will change the transport patterns of mercury not only in the atmosphere but

also between matrixes.

**Table**

**Table 1.** The correlation of the time air masses spent over different surfaces and GEM concentration shown for the different seasons (DJF, MAM, JJA and SON). Values for both R and $R^2$ are shown.

| $R^2$ | DJF | MAM | JJA | SON | | R | DJF | MAM | JJA | SON |
|---|---|---|---|---|---|---|---|---|---|---|
| Sea ice | 0.28 | 0.01 | 0.00 | 0.24 | | Sea ice | -0.53 | 0.11 | 0.04 | 0.49 |
| Snow | 0.17 | 0.00 | 0.06 | 0.03 | | Snow | 0.41 | -0.04 | -0.25 | -0.17 |
| Land | 0.03 | 0.02 | 0.02 | 0.44 | | Land | -0.17 | 0.17 | 0.12 | -0.66 |
| Sea | 0.22 | 0.03 | 0.14 | 0.32 | | Sea | 0.47 | -0.17 | 0.37 | -0.56 |

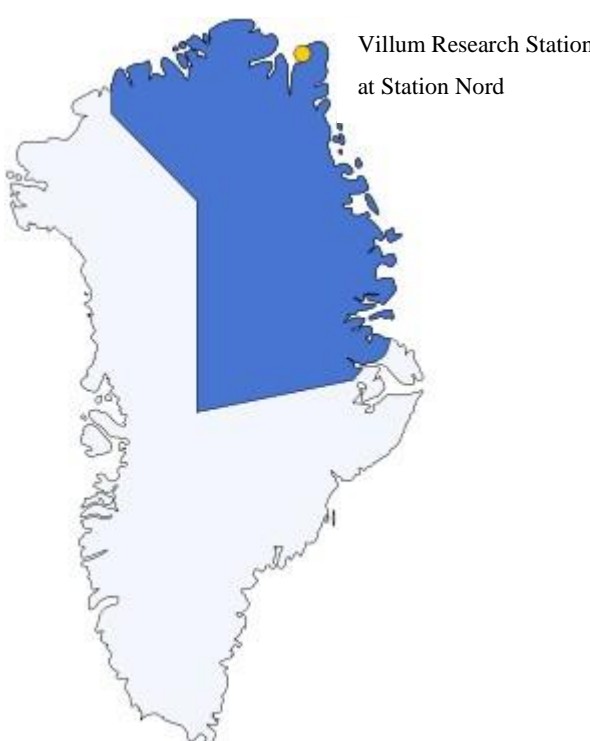

Villum Research Station
at Station Nord

**Figure 1.** The position of Villum Research Station at Station Nord in North Greenland. The blue area represents the Greenlandic National Park.

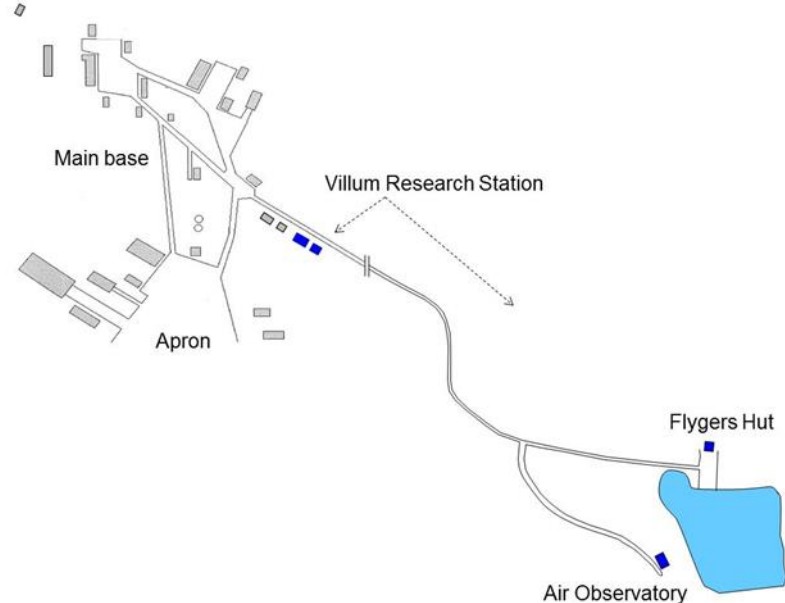


**Figure 2.** Map of Villum Research Station with its buildings (blue) relative to Station Nord military outpost. Flygers Hut and Air Observatory are located about 2 km outside main base of Station Nord. Until 2014 all measurements were performed in Flygers Hut, thereafter they were moved to Air Observatory.

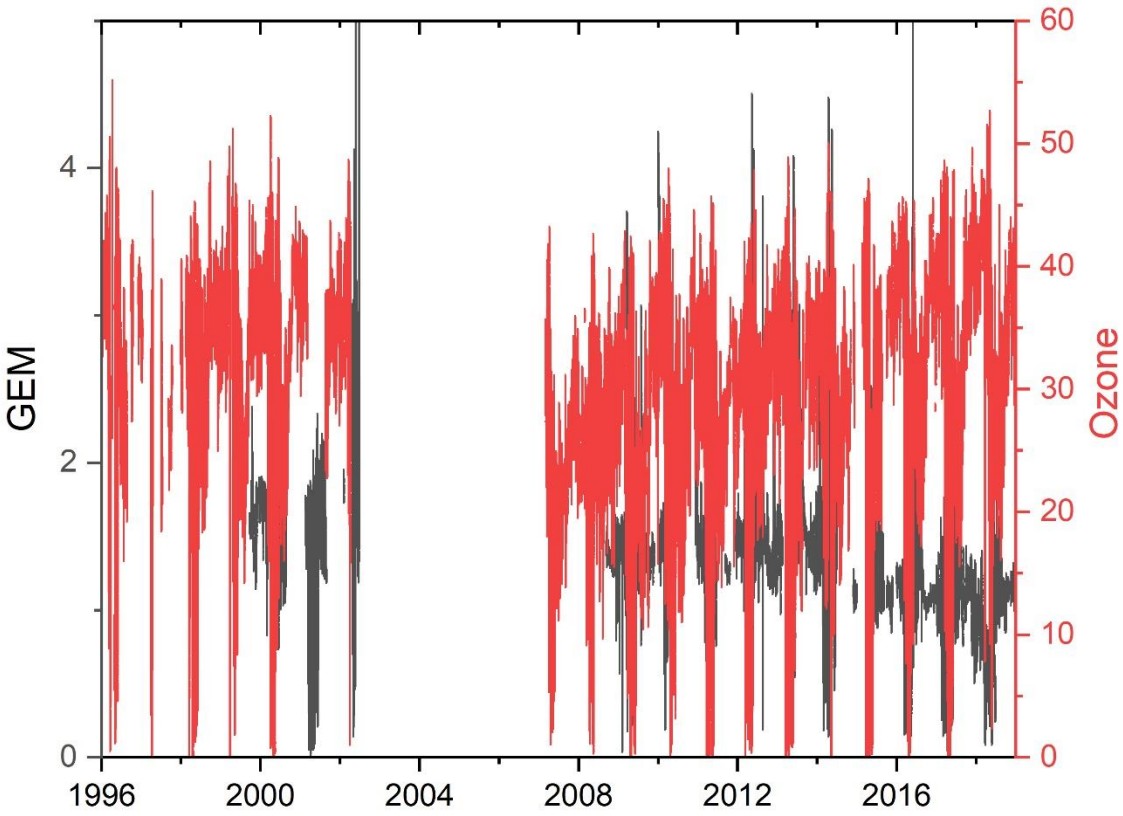

**Figure 3.** Time series of the concentration of GEM and the mixing ratio of ozone at Villum Research Station.

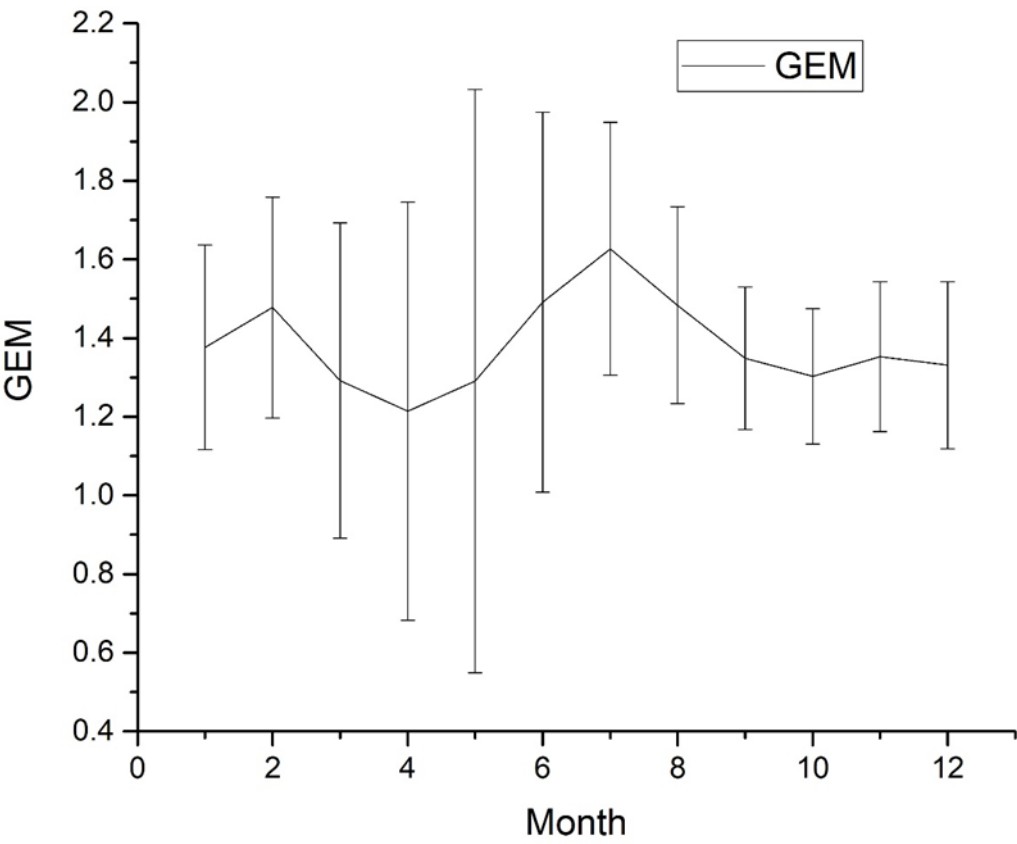


**Figure 4.** Monthly averages of GEM for the years 1999 to 2002 and 2008 to 2018 at Villum Research Station. The spread in monthly mean value is shown as plus/minus one std. dev.


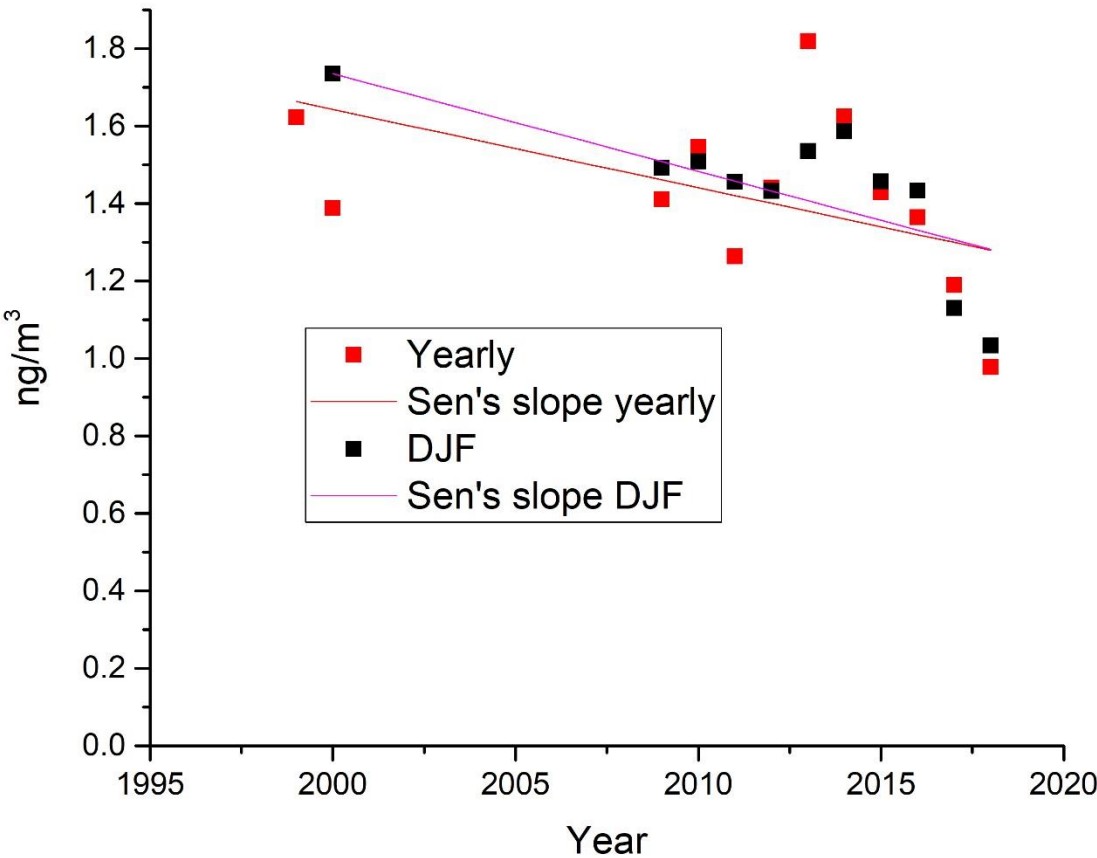

**Figure 5.** Yearly (Orange) and winter season (December-January-February, Blue) average values of measured GEM concentrations at Villum with trend lines.


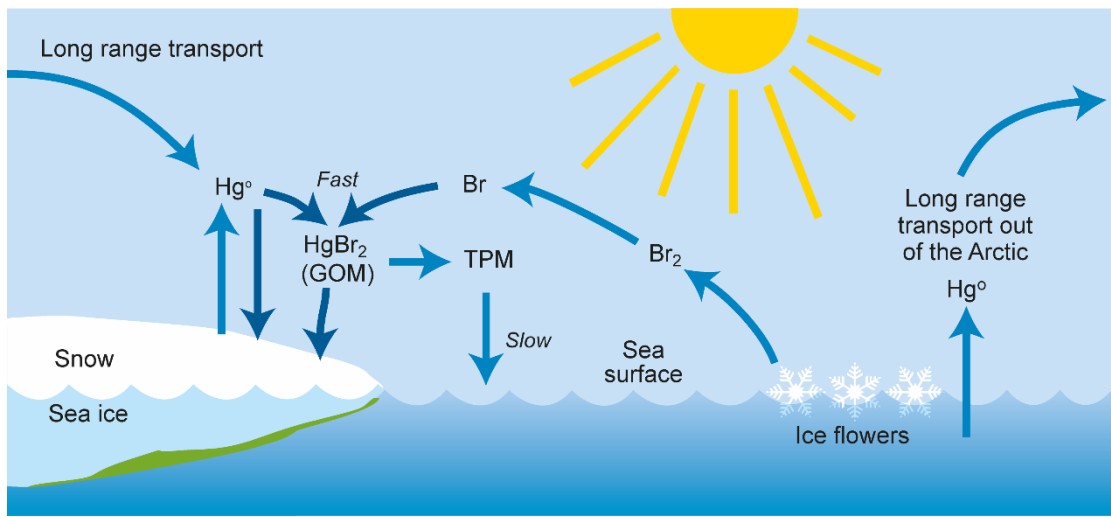

**Figure 6.** The mercury cycle in the Arctic atmosphere, where gaseous elemental mercury ($Hg^0$) is converted to gaseous oxidised mercury (GOM) that is quickly either deposited or converted into total particulate mercury (TPM). The chemical composition of GOM is unknown and $HgBr_2$ is one suggestion among many.

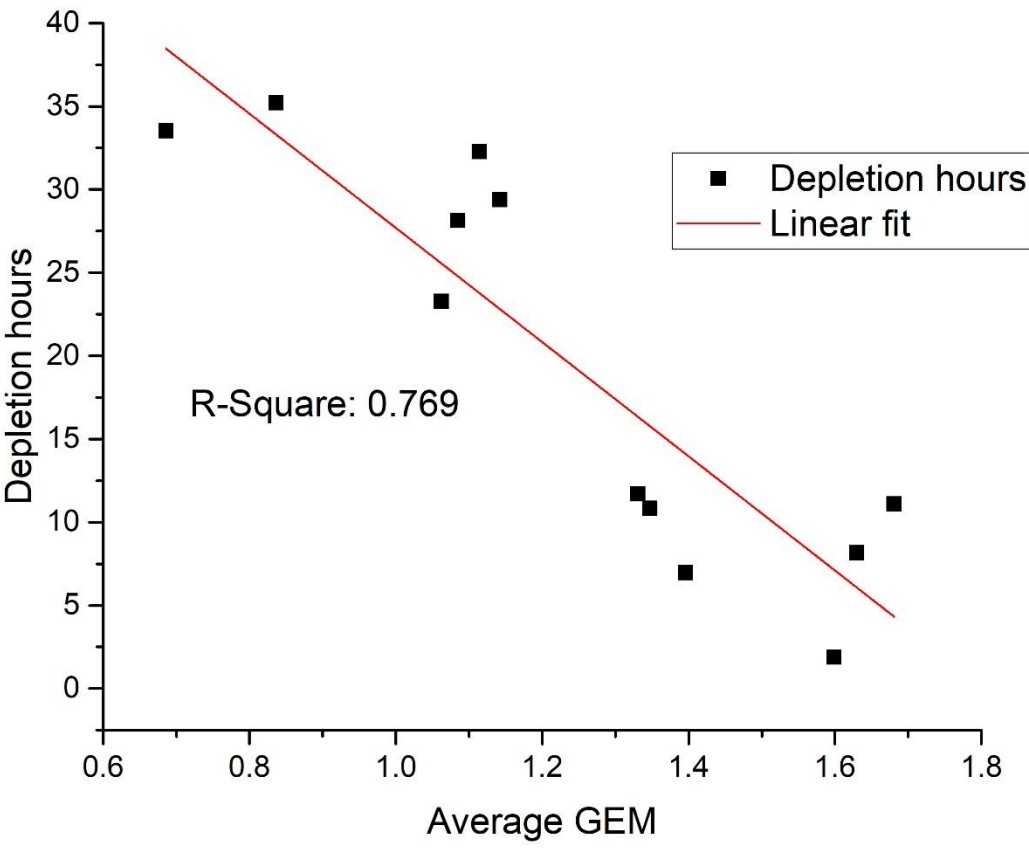

**Figure 7.** Frequency of depletion episodes versus average GEM concentration in March, April and May.


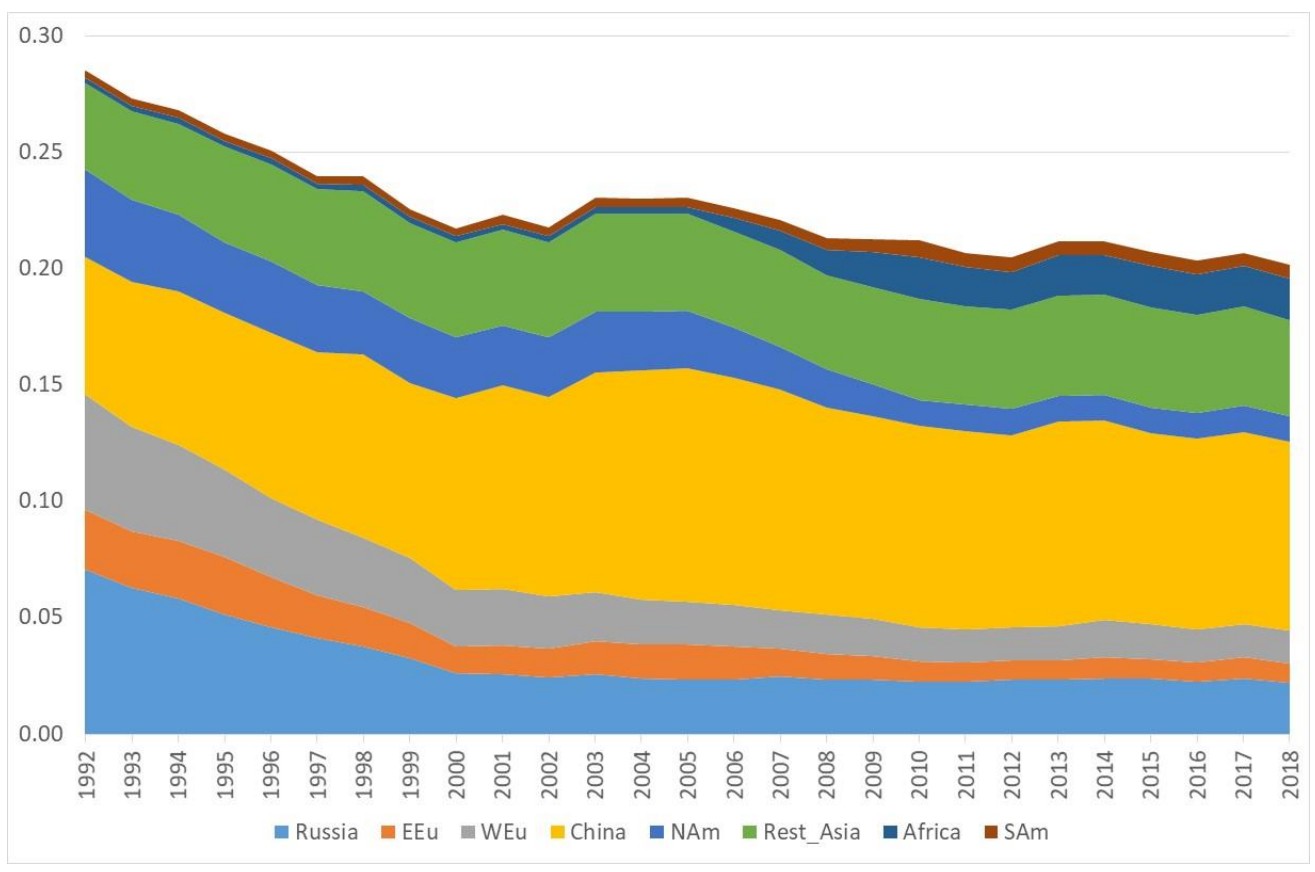

**Figure 8.** Model calculation with variable emissions of the source apportionment of the direct anthropogenic contribution to the annual average GEM concentrations at Villum. The DEHM model used two years (1990 and 1991) to spin up the model. Source regions: Russia = Russia; EEU = East Europe; WEu = West Europe; China = China; Africa = Africa; SAm = South America. Unit: ng m$^{-3}$.

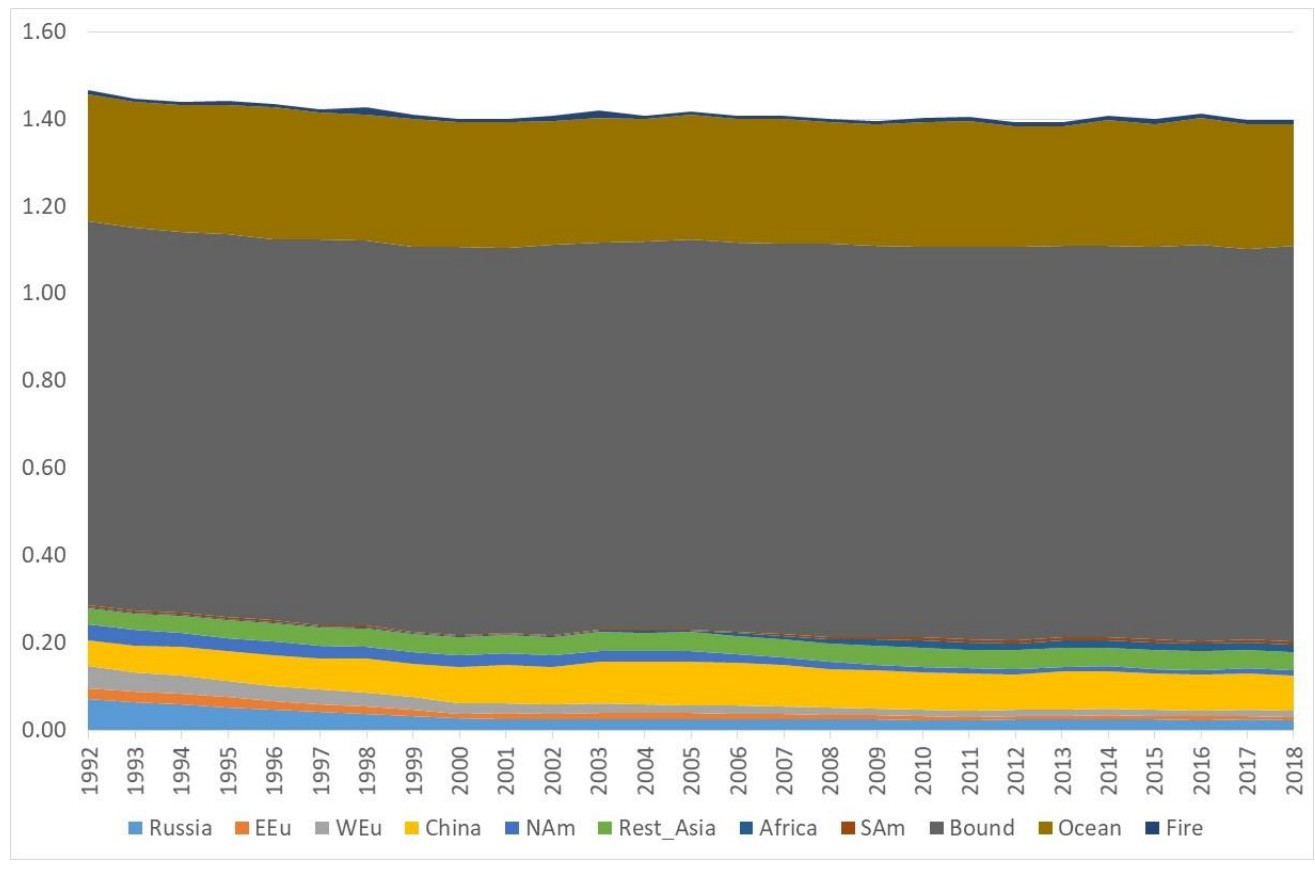


**Figure 9.** Model calculation with variable emissions of the source apportionment of annual average GEM at Villum. The DEHM model used two years (1990 and 1991) to spin up the model. In the model reemission from ocean and contribution from boundary conditions at equator included. Source regions: Russia = Russia; EEU = East Europe; WEu = West Europe; China = China; Africa = Africa; SAm = South America; Bound = Boundary Condition; Ocean = Ocean; Fire = Wildfire. Unit: ng m$^{-3}$.

## Author Contribution

All co-authors were involved in the scientific discussions of the paper

Henrik Skov: Project leader and principal writer

Jens Hjorth: Co-writer, coordination of statistical analysis

Bjarne Jensen: Calibration, tests and setup of instruments

Christel Christoffersen: Calibration, tests and setup of instruments

Maria Bech Poulsen: Trend analysis and analysis of the relation between ozone and GEM

Jesper Baldtzer Liisberg: Analysis of depletion events

David Beddows: Trajectory clustering analysis and K-statistics.

Manual Dall'Osto: Trajectory clustering analysis and K-statistics as well as overall design of article

Jesper Heile Christensen: Model calculations by DEHM.

**Acknowledgement**

This study was funded by the Danish Environmental Protection Agency (DANCEA funds for Environmental Support to the Arctic Region) and by the European program ERA-PLANET project iGOSP and iCUPE. The Royal Danish Air Force is acknowledged for providing free transport of equipment to Station Nord, and the staff at Station Nord is especially
acknowledged for excellent technical support. The Villum Foundation is gratefully acknowledged for financing the new research station, Villum Research Station. Daniel Charles Thomas and Jakob Boyd Pernov are acknowledged for their assistance with proofreading and language. The anonymous referees are acknowledged for their constructive comments.

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
