# Peer review of "Variability in Gaseous Elemental Mercury at Villum Research Station, Station Nord, in North Greenland from 1999 to 2017"

_Atmospheric Chemistry and Physics, 2019_

## Referee Comment (RC1) · Anonymous Referee #2 · 28 Apr 2020

This manuscript by Skov et al. presents a time-series of gaseous elemental mercury (GEM) concentrations at Villum Research Station (Station Nord), Greenland from 1999 to 2002 and 2009 to 2017. Alert (Canada), Station Nord, and Ny-Ålesund (Spitzbergen) are presently the only three long-term monitoring stations for GEM in the Arctic. As such, the data presented here are extremely valuable and I would like to acknowledge the authors for their work and dedication. That being said, I do not find the interpretation of the data convincing, mostly due to a confusing Material & Methods section (see suggestions below). The Results and Discussion section is also difficult to follow; I wasn't always sure whether the authors were referring to the annual or seasonal trend. Reorganizing the discussion per season could help. Finally, I think there is a lack of

sufficient references, especially recent ones. Below are some detailed comments and suggestions that will hopefully help the authors to improve their manuscript.

Measurements section

1. Line 89: "several generations of the instrument have been used (A, B, and X version)". Could you add somewhere (in the text and/or on Figure 3) the dates at which the Tekran instrument was changed? Given the 20% intercomparison uncertainty between two instruments (Slemr et al., 2015), that should I think be taken into account when performing a trend analysis. The winter trend seems driven by the high value in 2000 and the low value in 2017. Does it coincide with a different instrument being used? According to Angot et al. (2016), you used a Tekran 2537A at least from 2011 to 2015. According to Kamp et al. (2018), you used a Tekran 2537X in spring 2016. When did you switch? Did you measure GEM concentrations with the two instruments for a certain period of time in order to evaluate the intercomparison uncertainty? The lack of information casts doubts on the trend analysis. GEM trend analysis is of utmost importance for the effectiveness evaluation of the Minamata Convention. However, potential implication of the use of multiple instruments for GEM trend analysis is somewhat overlooked by the community. A discussion on the matter could strengthen the conclusions of the manuscript.

2. What is the time resolution of the GEM measurements? 5 or 15 minutes? Did you use the 5/15 min data for the trend analysis or hourly means/medians, or annual averages?

Trend analysis

How did you perform the trend analysis? Please describe the method in the Material and Methods Section. It seems that you are simply using the regression line. It is of common practice to use the Sen's slope and Mann-Kendall test for trend analysis (e.g., Berg et al., 2013; Cole and Steffen, 2010; Martin et al., 2017). Again, the lack of information casts doubts on the trend analysis.

Modeling section

1. Can you please provide more information regarding the Hg chemistry in the model (e.g., main oxidation and reduction pathways)?

2. Lines 126-130: I do not understand which Hg emissions were used in the simulations. You mention using "global historical AMAP Hg emissions inventories 1999 to 2017" which seems to suggest different emissions every year from 1999 to 2017. You later say that "the anthropogenic emissions are variable up to 2010 where after they are constant". Please clarify: Does this mean you used variable emissions from 1999 to 2010, then constant emissions (equal to 2010 emissions?) for 2010-2017? Please also clarify what is the reason for doing so. Are you trying to investigate the influence of changing anthropogenic emissions on GEM concentrations at Station Nord? I would like to see a Table summarizing which simulations were done in order to address which question/hypothesis, and a list of sensitivity simulations. See for instance Table 2 in Giang et al. (2018) or Table 2 in Travnikov et al. (2017).

3. Line 133: Can you please clarify what the "prescribed boundary conditions on 1.5 ng/m3" is? How did you come up with this value of 1.5 ng/m3? Is it based on a run with a global model? Or is it what you consider the Northern Hemisphere background concentration? If so, where does this value come from? I also do not understand why you refer to this value as a hemispheric background while it also represents "transport from sources in the southern hemisphere". This is confusing. In addition, can you please perform a sensitivity analysis here? Based on your simulations, direct anthropogenic transport only accounts for 14-17% of GEM. Does this direct anthropogenic influence changes if you decrease the prescribed boundary condition?

Line-by-line comments

Lines 42-43: "natural, anthropogenic, and reemission, accounting for roughly 10, 30, and 60% of the emissions, respectively". The authors should cite Amos et al. (2013) here.

Lines 43-51: Please discuss the most recent estimate prepared for the Global Mercury Assessment 2018 by Outridge et al. (2018). Anthropogenic emissions are ∼2500 tons per year, natural and secondary emissions ∼5500 tons per year, and evasion from ocean accounts for ∼60% of the sum of natural and secondary emissions.

Line 59: "evidence has been obtained from experimental and theoretical studies for a much shorter lifetime of GEM". What do you mean by "much shorter"? Please provide a range of values. Additionally, please considering adding Horowitz et al. (2017) to the list of references here.

Line 68-69: "Today China accounts for about 40% of the global Hg emission (Jiskra et al., 2018; Muntean et al., 2014)". Citing Jiskra et al. (2018) is not appropriate here. I suggest the following papers instead: Streets et al. (2019, 2018, 2017).

Line 70: "decline in the GEM concentration of between -1.5 and 2.2% per year (Obrist et al., 2018)". Citing Obrist et al. (2018) is not appropriate here. Please consider citing Zhang et al. (2016) instead.

Line 84: "2015 when the measurements were moved to the newly build Air Observatory". Figure 2 says 2014 (caption).

Lines 93-95: "the reproducibility for concentrations above 0.5 ng/m3 is within 20% based on parallel measurements with two Tekran 2537A mercury analyzers". Is it something you did as part of this study or are you referring to another study? If so, the reference is missing. You could cite Slemr et al. (2015) here.

Lines 154-156: "A seasonal pattern is observed for each year. In January and February, the level of ozone and GEM is rather stable. After the polar sunrise, the concentration starts to fluctuate strongly". It is pretty hard to see anything on Figure 3. Could you add a Figure giving the mean seasonal cycle of GEM (e.g. Figure 4 in Angot et al. (2016))?

Line 163: "A test of the importance of the value in 2000 . . .". How about the value in

2017?

Line 169: typo. Dommergue.

Lines 192-194: the end of that sentence is missing.

Lines 202-204: "Frequency of AMDE and GEM concentration in summer showed a poor negative correlation. If the deposited Hg during AMDE should be released again during snowmelt, a positive correlation would have been expected, but this was not observed". This hypothesis has already been tested, and the same conclusion reached, by Angot et al. (2016) based on data from Alert. "The increase of Hg(0) concentrations in summer could be due to the reemission of Hg deposited during springtime AMDEs. However, the comparison of the magnitude of the summer enhancement at Alert suggests otherwise". This is worth mentioning since the same conclusion is reached here.

Line 208: "The present study indicated that atmospheric input can be significant as well". I'm not sure I followed how you arrived at this conclusion.

Line 223: "The highest concentration of GEM was found in 2013". Isn't it in 2014 according to Figure 4?

Lines 225-226: "The DEHM model, using variable anthropogenic emissions as described above, shows a slightly decreasing concentration trend (see Figure 7)". I assume you mean Figure 8 here. Figure 7 shows the contribution of the various regions to the annual average GEM concentration at Villum. Also, you mention "variable anthropogenic emissions" while using constant emissions after 2010 according to the Material and Methods Section. If you observed a decreasing trend with constant emissions, that means the decreasing trend is not due to decreasing anthropogenic emissions. This modeling section is quite confusing, reason why I suggest major changes in the Material and Methods section to clarify what you are doing.

Line 234: missing parenthesis after Dibble et al., 2012.

Line 242: "in separate calculations with DEHM". What exactly did you do? Please

improve the Material and Methods section accordingly and provide a summary of all the simulations performed (and why).

Lines 242-257: Chen et al. (2018) also found that Hg deposition in the Arctic is mainly due to emissions from Asia. Additionally, the source-apportionment analysis performed with four global models for the Global Mercury Assessment 2018 shows that the Arctic is predominantly influenced by long-range transport from East Asia.

Lines 259-264: How do you results compare to the study by Dastoor et al. (2015): the authors found that changes in meteorology and decline in emissions in North America and Europe contribute equally to the decrease in surface air Hg(0) concentrations. Additionally, you mention here a simulation where emissions are kept constant at the 2005 level, while meteorology is varying. Please clarify this in the list of simulations in the Material and Methods.

Lines 265-278: this entire paragraph is unclear because Table 2 is missing.

Line 278: "The DEHM model predicts that there is a maximum in GEM. . .". Could you please provide a direct comparison of observed vs. modeled time-series?

Lines 279-281: "In fact, the highest concentration of GEM is observed during summer and is attributed to release of GEM from the melting snow and ice pack from Hg deposited during AMDEs in spring". I do not understand. You say earlier in the manuscript that springtime AMDEs cannot explain the summertime enhancement: "Frequency of AMDE and GEM concentration in summer showed a poor negative correlation. If the deposited Hg during AMDE should be released again during snowmelt, a positive correlation would have been expected, but this was not observed".

Line 288: "Simulations of the concentrations at Villum using the DEHM model using fixed emission inventory show no significant trends". You say earlier that the simulations give a decreasing trend: "shows a slight decreasing concentration trend, -0.7% per year". Or, by "slight", do you mean that this trend is not significant?

[Figure]

References

Amos, H.M., Jacob, D.J., Streets, D.G., Sunderland, E.M., 2013. Legacy impacts of all-time anthropogenic emissions on the global mercury cycle. Global Biogeochem. Cycles 27, 410–421. https://doi.org/10.1002/gbc.20040

Angot, H., Dastoor, A., De Simone, F., Gårdfeldt, K., Gencarelli, C.N., Hedgecock, I.M., Langer, S., Magand, O., Mastromonaco, M.N., Nordstrøm, C., Pfaffhuber, K.A., Pirrone, N., Ryjkov, A., Selin, N.E., Skov, H., Song, S., Sprovieri, F., Steffen, A., Toyota, K., Travnikov, O., Yang, X., Dommergue, A., 2016. Chemical cycling and deposition of atmospheric mercury in polar regions: review of recent measurements and comparison with models. Atmos. Chem. Phys. 16, 10735–10763. https://doi.org/10.5194/acp-16-10735-2016

Berg, T., Pfaffhuber, K.A., Cole, A.S., Engelsen, O., Steffen, A., 2013. Ten-year trends in atmospheric mercury concentrations, meteorological effects and climate variables at Zeppelin, Ny-Ålesund. Atmospheric Chemistry and Physics 13, 6575–6586. https://doi.org/10.5194/acp-13-6575-2013

Chen, L., Zhang, W., Zhang, Y., Tong, Y., Liu, M., Wang, H., Xie, H., Wang, X., 2018. Historical and future trends in global source-receptor relationships of mercury. Science of The Total Environment 610, 24–31. https://doi.org/10.1016/j.scitotenv.2017.07.182

Cole, A.S., Steffen, A., 2010. Trends in long-term gaseous mercury observations in the Arctic and effects of temperature and other atmospheric conditions. Atmos. Chem. Phys. 10, 4661–4672. https://doi.org/10.5194/acp-10-4661-2010

Dastoor, A., Ryzhkov, A., Durnford, D., Lehnherr, I., Steffen, A., Morrison, H., 2015. Atmospheric mercury in the Canadian Arctic. Part II: Insight from modeling. Science of The Total Environment, Special Issue: Mercury in Canada's North 509–510, 16–27. https://doi.org/10.1016/j.scitotenv.2014.10.112

Giang, A., Song, S., Muntean, M., Janssens-Maenhout, G., Harvey, A., Berg, E., Eckley

Selin, N., 2018. Understanding factors influencing the detection of mercury policies in modelled Laurentian Great Lakes wet deposition. Environmental Science: Processes & Impacts. https://doi.org/10.1039/C8EM00268A

Horowitz, H.M., Jacob, D.J., Zhang, Y., Dibble, T.S., Slemr, F., Amos, H.M., Schmidt, J.A., Corbitt, E.S., Marais, E.A., Sunderland, E.M., 2017. A new mechanism for atmospheric mercury redox chemistry: implications for the global mercury budget. Atmos. Chem. Phys. 17, 6353–6371. https://doi.org/10.5194/acp-17-6353-2017

Jiskra, M., Sonke, J.E., Obrist, D., Bieser, J., Ebinghaus, R., Myhre, C.L., Pfaffhuber, K.A., Wängberg, I., Kyllönen, K., Worthy, D., Martin, L.G., Labuschagne, C., Mkololo, T., Ramonet, M., Magand, O., Dommergue, A., 2018. A vegetation control on seasonal variations in global atmospheric mercury concentrations. Nature Geoscience 11, 244–250. https://doi.org/10.1038/s41561-018-0078-8

Kamp, J., Skov, H., Jensen, B., Sørensen, L.L., 2018. Fluxes of gaseous elemental mercury (GEM) in the High Arctic during atmospheric mercury depletion events (AMDEs). Atmos. Chem. Phys. 18, 6923–6938. https://doi.org/10.5194/acp-18-6923-2018

Martin, L.G., Labuschagne, C., Brunke, E.-G., Weigelt, A., Ebinghaus, R., Slemr, F., 2017. Trend of atmospheric mercury concentrations at Cape Point for 1995–2004 and since 2007. Atmos. Chem. Phys. 17, 2393–2399. https://doi.org/10.5194/acp-17-2393-2017

Muntean, M., Janssens-Maenhout, G., Song, S., Selin, N.E., Olivier, J.G.J., Guizzardi, D., Maas, R., Dentener, F., 2014. Trend analysis from 1970 to 2008 and model evaluation of EDGARv4 global gridded anthropogenic mercury emissions. Science of The Total Environment 494–495, 337–350. https://doi.org/10.1016/j.scitotenv.2014.06.014

Obrist, D., Kirk, J.L., Zhang, L., Sunderland, E.M., Jiskra, M., Selin, N.E., 2018. A review of global environmental mercury processes in response to human and natural

perturbations: Changes of emissions, climate, and land use. Ambio 47, 116–140. https://doi.org/10.1007/s13280-017-1004-9

Outridge, P.M., Mason, R.P., Wang, F., Guerrero, S., Heimbürger-Boavida, L.E., 2018. Updated Global and Oceanic Mercury Budgets for the United Nations Global Mercury Assessment 2018. Environ. Sci. Technol. 52, 11466–11477. https://doi.org/10.1021/acs.est.8b01246

Slemr, F., Angot, H., Dommergue, A., Magand, O., Barret, M., Weigelt, A., Ebinghaus, R., Brunke, E.-G., Pfaffhuber, K.A., Edwards, G., Howard, D., Powell, J., Keywood, M., Wang, F., 2015. Comparison of mercury concentrations measured at several sites in the Southern Hemisphere. Atmos. Chem. Phys. 15, 3125–3133. https://doi.org/10.5194/acp-15-3125-2015

Streets, D.G., Horowitz, H.M., Jacob, D.J., Lu, Z., Levin, L., ter Schure, A.F.H., Sunderland, E.M., 2017. Total Mercury Released to the Environment by Human Activities. Environ. Sci. Technol. 51, 5969–5977. https://doi.org/10.1021/acs.est.7b00451

Streets, D.G., Horowitz, H.M., Lu, Z., Levin, L., Thackray, C.P., Sunderland, E.M., 2019. Five hundred years of anthropogenic mercury: spatial and temporal release profiles. Environ. Res. Lett. 14, 084004. https://doi.org/10.1088/1748-9326/ab281f

Streets, D.G., Lu, Z., Levin, L., ter Schure, A.F.H., Sunderland, E.M., 2018. Historical releases of mercury to air, land, and water from coal combustion. Science of The Total Environment 615, 131–140. https://doi.org/10.1016/j.scitotenv.2017.09.207

Travnikov, O., Angot, H., Artaxo, P., Bencardino, M., Bieser, J., D'Amore, F., Dastoor, A., De Simone, F., Diéguez, M.D.C., Dommergue, A., Ebinghaus, R., Feng, X.B., Gencarelli, C.N., Hedgecock, I.M., Magand, O., Martin, L., Matthias, V., Mashyanov, N., Pirrone, N., Ramachandran, R., Read, K.A., Ryjkov, A., Selin, N.E., Sena, F., Song, S., Sprovieri, F., Wip, D., Wängberg, I., Yang, X., 2017. Multi-model study of mercury dispersion in the atmosphere: atmospheric processes and model evaluation. Atmos.

[Figure]

Chem. Phys. 17, 5271–5295. https://doi.org/10.5194/acp-17-5271-2017

Zhang, Y., Jacob, D.J., Horowitz, H.M., Chen, L., Amos, H.M., Krabbenhoft, D.P., Slemr, F., Louis, V.L.S., Sunderland, E.M., 2016. Observed decrease in atmospheric mercury explained by global decline in anthropogenic emissions. PNAS 113, 526–531. https://doi.org/10.1073/pnas.1516312113
* * *

---

## Referee Comment (RC2) · Anonymous Referee #1 · 29 Jun 2020

The manuscript by Skov et al. deals with an 11-yrs data set of atmospheric GEM observations in the high Arctic. As such, it is meriting for publication in ACP. My rating for scientific significance is excellent and scientific quality is good while the presentation is of only low to fair quality. I generally agree with the comments put by Referee 2 in his/her exhaustive review.

Two items that disturbed me are the substandard quality of graphs and the lack of updated information on Polar Hg cycling.

Following the authors' reply to referee 2, I subjectively judge that the authors have responded to the remarks in an adequate way (presentation quality raised to "good") .

I have a few remaining comments concerning previous Figure 5 and the atmospheric/snow chemistry:

There are substantial advances in the knowledge of Arctic Hg cycling using stable isotopes.

- An example is that dry deposited $Hg^0$ rather than AMDE-sourced Hg comprises the majority (~76–91%) of snowmelt $Hg^{II}$ in the coastal Arctic [1, 2].

It is appropriate that this info is added to Fig. 5. and mentioned in Section 3. 1

There has also been a progression in the assessment of Br-induced GEM oxidation.

- The authors should consider responding and citing e. g. the following paper: Wang S, McNamara SM, Moore CW, Obrist D, Steffen A, Shepson PB, et al. Direct detection of atmospheric atomic bromine leading to mercury and ozone depletion. Proceedings of the National Academy of Sciences 2019; 116: 14479-14484.

References

1.    Douglas, T.A., and J.D. Blum, *Mercury Isotopes Reveal Atmospheric Gaseous Mercury Deposition Directly to the Arctic Coastal Snowpack.* Environmental Science & Technology Letters, 2019. **6**(4): p. 235-242.
2.    Jiskra, M., et al., *Insights from mercury stable isotopes on terrestrial–atmosphere exchange of Hg(0) in the Arctic tundra.* Biogeosciences, 2019. **16**(20): p. 4051-4064.

---

## Author Response (AR3)

**Reply to referees and editor**

**Reply to referee 1**

All replies to referee questions and comments are in plain text and all changes to the text of the manuscript are in inverted

5    commas (""). For clarity, we list the question in *italics*. Furthermore, the graphs have been updated using Origin instead of

Excel, as was a comment by both referee 1 and 2.

We thank the reviewer for a precise review of the article. We have answered the questions, added, and modified text when

necessary. More references have been added as well.

10   *There are substantial advances in the knowledge of Arctic Hg cycling using stable isotopes.*

*• An example is that dry deposited Hg0 rather than AMDE-sourced Hg comprises the majority (~76–91%) of snowmelt HgII in*

*the coastal Arctic [1, 2].*

The two references have been added in the discussion and the following text added:

15   New Lines 249-252 "From studies of mercury isotopes at Utqiaġvik at the North coast of Alaska (Douglas et al., 2019) and

Toolik Research Station in central Alaska (Jiskra et al. 2019), it was found that most mercury in melt water was from

deposition of GEM and that a large majority of deposited oxidised mercury during AMDE was reduced and reemitted. Further,

studies are needed to determine if these results are valid also for more northern Arctic locations as Alert, Villum or Zeppelin."

Figure 5 old version now Figure 6 has been updates to include also deposition of GEM.

*There has also been a progression in the assessment of Br-induced GEM oxidation.*

*• The authors should consider responding and citing e. g. the following paper: Wang S, McNamara SM, Moore CW, Obrist D,*

*Steffen A, Shepson PB, et al. Direct detection of atmospheric atomic bromine leading to mercury and ozone depletion.*

*Proceedings of the National Academy of Sciences 2019; 116: 14479-14484.*

25   The Article has been included in the discussion as.

New Lines 232-235 "Recently, the Br induced oxidation of $Hg^0$ has been proven directly in a study, where Br, BrO, $O_3$, GEM

and RGM were measured simultaneously during AMDE and ODE and using a multiphase box model to study the complex set

of processes (Wang et al., 2019)."

**Reply to referee 2**

35 All replies to referee questions and comments are in plain text and all changes to the text of the manuscript are in inverted commas (""). Furthermore, the graphs have been updated using Origin instead of Excel.

We thank the reviewer for a thorough review of the article. We have answered all questions, added, and modified text when deemed necessary. Especially, we have modified the Material & Methods section and the Results and Discussion section to make them more straightforward to understand and explain better the data interpretation. More references have been

40 added as well.

The indicated lines are referring to the revised manuscript and the numbers in parenthesis are referring to lines in the ACPD version.

1. Line 91 (Line 89): We have rechecked, which instruments were in use during the period of the measurements. From, 1999 to 2002 only A model instruments were in use and they were applied again from 2009 to $1^{st}$ Dec.

45 2016, where they were replaced by a B model that was used until $3^{rd}$ Dec. 2017. The last month of 2017 and in 2018 X models were used. The referee states that the 20% uncertainty should be included into the uncertainty especially in the trend analyses. This uncertainty is a random uncertainty. All instruments are calibrated towards the same standard (vapour pressure of Hg using instrument Tekran 2505 calibration unit) and this preclude a systematic error. An explanation is added in line 90. We do trend analysis of yearly or seasonally averaged values

50 and thus the random uncertainty is minimized. If there is a systematic error, we correct for it following ISO Guide 98-3:2008 Uncertainty of measurement — Part 3: Guide to the expression of uncertainty in measurement (GUM:1995) and include this correction into the uncertainty. In Kamp et al. 2019, we used a separate setup based on two Tekran 2537X instruments and the measurements were independent of our monitoring activities that provide the result for this article.

55 Line (91-93): The sentence has been modified to make it clearer: "Several generations of the instrument have been used (A, B and X versions) but we estimate that the uncertainty of measuring GEM has remained unchanged during the years as they are all calibrated towards the same standard based on the vapour pressure of $Hg^0$ using Tekran 2505 calibration unit."

60 2. The first years 1999-2015 we used 5 minutes sampling. Thereafter we changed to 15 minutes sampling in order to decrease the consumption of Ar.

Trend analysis

In the trend analysis, we used yearly and seasonal mean values (3 months). Following the advice of Referee 2, in the revised

65 version of the manuscript we have used the non-parametric Mann-Kendahl test and Sen's slope calculation, instead of the

classical regression analysis that we applied in the manuscript under review, because of the advantages of this approach (no assumptions about the distribution of the measurements, low sensitivity to outlier values). The following text has been added to the 'Experimental Section':

Line 116-117 (108-109): "The calculation of inter annual trends were performed applying the non-parametric Mann-Kendahl test and Sens slope calculation, using the program developed by Salmi et al. (2002)."

The new trend analysis, including the GEM measurements from 2018 that have now been quality assured, show significant negative trends (at a 90% confidence level) of the autumn and winter (SON, DJF) average values but no significant trend of the annual averages (as in the previous analysis). We have decided not to report non-significant trends, thus we have omitted the previous Table 1. The initial part of the discussion of trends, starting by the beginning of Section 3 has been revised and is now the following:

Line 184-202 (180-193): "The measurements of GEM and ozone from 1996 to 2018 are shown in Figure 3. A seasonal pattern is observed for each year. In January and February, the level of ozone and GEM is rather stable. After the polar sunrise, the concentration starts to fluctuate strongly and ozone and GEM are depleted fast (during 2 to 10 hours). Figure 4 shows the variations of the yearly average GEM concentration and the average for the winter season between 1999 and 2018, where only periods with more than 50% data coverage have been included. The annual averages show a negative trend, however not significant at a 90% confidence level. The autumn (September-October-November) and the winter (December-January-February) season show both negative trends that are significant at a 90% confidence level (annual and winter data are shown in Figure 5). The trends, in percentage of the average GEM concentrations during these periods, are -1.7%/yr for the winter period and -1.4%/yr for the autumn. The annual trend remains non-significant also when excluding the years 1999 and 2000 or the extreme value in 2017. The lack of a significant annual trend seems to be explained by the high variability of the concentrations during the spring period as well as the fact that the GEM concentration during the summer period shows no evidence of a decreasing trend.

This result is similar to the result......"

Modelling section

1. The focus on this paper is the direct transport of GEM from sources to the measurements site. Therefore, we used simple first order chemistry as written in line 156-157.

2. The emission inventories applied have been clarified in the text, see line 131-137 (126-130):

"The global historical AMAP Hg emissions inventories for 1990-2010 have been used as the anthropogenic emissions (UNEP 2013) for the model run with variable emissions. The 1990 emissions have been used for the model calculations for the period 1990-1992, 1995 emissions for the years 1993-1997, 2000 emissions for 1998-2002, 2005 emissions for 2003-2007 and finally the 2010 emission for 2008-2017. The emissions for 2005 were used for the model run with constant emissions.

Emissions of mercury from biomass burning were based on CO emissions obtained from Global Fire Emissions Database, Version 3, (van der Werf et al. 2006; Van der Werf et al., 2003), where a fixed $Hg^0$/CO ratio of $8\times10^{-7}$ kg $Hg^0$/kg CO was applied. Emissions from oceans are based on calculated fluxes from the GEOS-Chem model (Soerensen et al. 2010)."

3.   Line 146-161 (133): The text have been modified in order to clarify the meaning of boundary condition. The model calculations is actually a sensitivity study, where the contributions from different sources as function of first order lifetime of GEM are estimated. Moreover, because it is a linear first order lifetime, it is quite easy to scale the different source areas including boundary conditions. The direct anthropogenic influence will be changed (be larger in percent) if the prescribed boundary conditions are decreased.

"The system has been set up with 11 different GEM tracers, which represent eight different anthropogenic source areas (Russia, Eastern Europe, Western Europe, China, North America, Rest of Asia, Africa and South America), biomass burning, ocean sources and the prescribed boundary conditions of 1.5 ng/m$^3$ for the entire period. The latter is introduced because of the long lifetime of $Hg^0$ and accounts for the transport across equator with the exchange velocity between the two hemispheres of about 1 year. The boundary condition concentration of 1.5 ng/m$^3$ represents the typical global background concentrations, which account for all emissions in both hemispheres, and are close to the concentrations at equator as given in Selin et al (2008). The boundary conditions were kept constant during the period covered by the model.

There have been made 2x3 different model runs covering the period from 1990 to 2017, with two main emissions setup, which are with either constant anthropogenic missions (using the emissions in 2005 for all years) or the variable emissions for 1990-2010. Each emissions setup is run with a simple fixed first order reaction lifetime for $Hg^0$ of 1 month, 6 months and 1 year, respectively. The model does not include Arctic mercury depletion in the runs presented here; it focuses only on the direct long-range transported mercury contribution to the GEM concentration at Villum. For each model run the contributions of the 11 different tracers are estimated in order to investigate this contribution as function of the fixed first order reaction lifetime for $Hg^0$, changing meteorology and changing emissions."

Lines 43-48 (43-51): An update of the text has been made and inserted

"The sources of mercury in the environment can be divided into natural, anthropogenic, and reemission, accounting for 2.1, 2.5 and 3.4 ktonnes of the emissions, respectively (Outridge et al. 2018). This is in good agreement with other estimates. The global anthropogenic emissions of mercury were estimated as 2.5 ktonnes in 2010 (UNEP 2013; AMAP/UNEP 2013) and including the large uncertainty on these numbers, they are not significantly different. According to an estimate by (Pirrone et al. 2010) natural sources and reemission processes (hereafter referred to as ´background sources´), accounted for 5207 Mg per year in 2005 while the amount of new anthropogenic inputs is 2320 Mg per year also close to the latest emission estimate (Outridge et al. 2018)."

Lines (225-226): (Previous Figure 7 and 8 now) Figure 8 and 9. Both Figure 8 and Figure 9 show decreasing concentrations but it more clearly seen in Fig 9.  In the first draft (in ACPD) the model output with constant emissions were shown. This is

now corrected and the model results presented in Figure 8 and 9 are with variable emissions.  As reply to reviewer, the use of variable emissions has been clarified in the figure captions:

"Figure 8: Model calculation with variable emissions of the source apportionment of the direct anthropogenic contribution to the annual average GEM concentrations at Villum. The DEHM model used two years (1990 and 1991) to spin up the model. Source regions: Russia = Russia; EEU = East Europe; WEu = West Europe; China = China; Africa = Africa; Sam = South America. Unit: ng m$^{-3}$."

"Figure 9: Model calculation with variable emissions of the source apportionment of annual average GEM at Villum. The DEHM model used two years (1990 and 1991) to spin up the model. In the model reemission from ocean and contribution from boundary conditions at equator included. Source regions: Russia = Russia; EEU = East Europe; WEu = West Europe; China = China; Africa = Africa; Sam = South America; Bound = Boundary  Condition; Ocean = Ocean; Fire = Wildfire. Unit: ng m$^{-3}$."

Line 283 (242): It was not a separate calculation. The text has been corrected for this:

"In the model calculations with DEHM, it was found that emissions from China had larger relative importance during the summer than in the winter season."

Lines 283-291 (259-264): We have included a comparison with the results from Dastoor et al (2015) in the new version of the text:

New text:

"Results obtained by applying the DEHM model to simulate GEM concentrations at Villum indicate that changes in the direct atmospheric transport from source areas to Villum cannot explain the observed trend. We have found that the simulated yearly and seasonal GEM values show very little variability and no significant trend over the years 2000-2015, when the emission sources are kept constant at the 2005 level while the meteorology is varying and treated as described above. That is opposite to results by Dastoor et al. (2015) for model run with constant emissions. The main reason for that is perhaps that processes as chemistry and surface exchanges in Dastoor et al. (2015), are more depending on the atmosphere and surface conditions than the simple setup in the present version of DEHM.  There are better agreement between our results and Dastoor et al (2015) for the model setup with variable emissions. We see a decrease of 0.08 ng/m$^3$ between 1992 and 2005 for Villum, while Dastoor et al. found approximately 0.1 ng/m$^3$.  The study by Hirdman et al. (2010) of long term trends of sulfate and BC in the Arctic also concludes that changes in atmospheric transport only can explain a small fraction (0.3-7.2%) of the observed trends."

Line-by-line comments.

165     Lines (42-43): An update of the text has been made see the answer above.

        Lines (43-51): We were not aware of the Outridge et al. paper. We are referring to this one as primary reference for the
        discussion of $Hg^0$ emissions to the atmosphere. See earlier

170
        Line 59-66 (59): The sentence is replaced by:
        "The atmospheric lifetime of GEM has earlier been estimated to be in the range of about one year (Steffen et al. 2008), while
        those of oxidized forms of mercury are shorter. Theoretical and laboratory studies showed that the lifetime of GEM towards
        Br initiated oxidation is much shorter than 1 year (Goodsite et al. 2004, 2012, Donohoue et al. 2006, Dibble et al. 2012,
175     Balabanov et al. 2005, Jiao and Dibble 2017; Donohoue et al. 2005). Applying the latest kinetic data, Horowitch et al. (2017)
        found a lifetime in the atmosphere of GEM against oxidation of 2.7 months using the GEOS-CHEM model cobbled to an ocean
        general circulation model (MITgcm).  Including photoreduction, the lifetime of total gaseous mercury (TGM) was found to be
        5.2 months close to the value 6.1 months of Holmes et al. 2010 but applying a much higher Br concentration and thus also a
        faster photoreduction."
180
        Line 79 (68-69): The references have been changed from Jiska et al. 2018 to Streets et al. (2019, 2018, 2017).

        Line 80 (70): The references have been changed from Obrist et al. (2018) to Zhang et al. (2016).

185     Line 94 (84): Correction in the text has been made as it was in 2014 that the monitors were moved.

        Lines (93-95): The 20% were determined in Skov et al. 2004 and confirmed in the present work.

190     Lines 185 (154-156): A new figure (Figure 4) is added showing the seasonal variation:
        Figure Caption: "Figure 4 Monthly averages of GEM for the years 1999 to 2002 and 2008 to 2018. The whiskers show the ± 1
        std. dev. of the monthly averages."

        Line (163): We have added results from 2018 as the final quality control is finished

195     Line 205 (169): Corrected

        Lines 228-231 (192-194): Sentence is now completed.
        "An important point for the parameterization of GEM depletion is that bromine induced atmospheric mercury depletion
        event (AMDE) often was observed under stagnant wind conditions and not only during situations with strong wind that may
200     cause bromine release as proposed earlier (see Yang et al. 2020)."

        Lines 241-245 (202-204): The discussion has been extended:
        "In fact, the analyses indicate that AMDE is a net sink for mercury, which is in agreement with direct flux measurements
205     (Brooks et al. 2006). Interestingly, (Angot et al. (2016) found a positive feedback between AMDE in spring and the
        concentration of GEM in summer at Alert that was attributed to reemission of mercury. Contrary to this result, even the

annual mean value at Villum had a negative correlation with AMDE hours. Though this correlation is weak, it is an indication that AMDEs affect the GEM concentration level at Villum and represent a net sink for GEM."

Line 247-251 (208): The sentence has been changed:

"The present study indicates that there is an atmospheric input as well. The significance of this source depends on its chemical form. Previously atmospheric deposited mercury has been identified to be bioavailable (Moller et al. 2011) and thus might still be dominant for the mercury found in the Arctic foodweb?"

Line 263 (223): No the yearly average is higher in 2013 than in 2014. The average concentration during winter is highest in 2013. This is now specified in the text.

Lines 267 (225-226): See corrections above where Figure captions have been extended to explain that Figure 7 shows only the direct contribution from anthropogenic sources, whereas Figure 8 shows the total contribution to GEM .

Line 275 (234): Missing parenthesis is added

Line 283 (242): The word "separate" has been deleted in sentence

Lines 283 (242-257): The following paragraph has been extended: "In the calculations with DEHM, it was found that emissions from China had larger relative importance during the summer than in the winter season; however, this difference was only significant for relatively short (less than 1 year) atmospheric lifetimes of GEM. The calculations for Villum were performed for the year 2001. This result agrees with Chen et al. (2018), who found that East Asia is the main source for mercury deposition in Arctic. Similar result is also reported by AMAP (AMAP 2018). Durnford et al. (2010), applying the GRAHM model, investigated the contribution of different source regions to total mercury as well as GEM concentrations at several Arctic monitoring stations at different seasons of the year. They found that for the yearly concentration averages and their variability at the Arctic stations, including Villum, Asian emissions were the most important, accounting for more than the sum of the contributions from Europe, Russia and North America."

Lines (259-264): This point was addressed in the model section above

Lines 325-337 (265-278): The missing Table is inserted and discussion checked.

Line (278): It does not give any meaning to compare measurements and modelled results with the current version of DEHM looking at shorter time scales. DEHM ran with a constant first order lifetime and thus the short-term variation of modelled GEM concentrations are only due to transport, whereas measured GEM is dependent on transport, chemistry and other processes. DEHM can thus only be applied here to say something about yearly average concentration and trends based on the lifetime, emissions and transport. The discussion of seasonality in measurements and model results has been removed

Lines (279-281): The sentence has been deleted, as it is confusing.

Line (288): The decreasing trend of -0.7% is as written in the text for the model run with variable emissions, while this paragraph is for the model run with fixed emissions as also explained in the text, e.g. the variability of the transport patterns does not give any significant trend.

**Reply to editor**

Dear Editor

Thank you for these last comments. Under each question from editor, I have added the answer in red.

Editor Decision: Publish subject to minor revisions (review by editor) (01 Sep 2020) by Ashu Dastoor

Comments to the Author:

Please revise the manuscript according to my comments below. Thank you

Non-public comments to the Author:

1. Please remove "The" at the beginning and add a comma after Station Nord in the title of paper:

Variability in Gaseous Elemental Mercury at Villum Research Station, Station Nord, in North Greenland from 1999 to 2017

This has been done in the text.

2. Please correct first sentence, second paragraph, in the introduction to:

The sources of mercury in the environment can be divided into terrestrial emissions (including geogenic, biomass burning and reemissions from soils and vegetation), anthropogenic and oceanic evasion, accounting for roughly 2.1, 2.5 and 3.4 ktonnes of the emissions, respectively (Outridge et al. 2018).

We have changed the text but anthropogenic evasion is not the right term, so instead is add;

"The sources of mercury in the environment can be divided into natural, terrestrial emissions (including geogenic, biomass burning and reemissions from soils and vegetation), anthropogenic, and oceanic emissions, accounting for 2.1, 2.5 and 3.4 ktonnes of the emissions, respectively (Outridge et al. 2018)."

3. Please correct the last statement, page 1, in introduction to:

Applying the latest kinetic data, Horowitz et al. (2017) found lifetime of GEM in the atmosphere with respect to gaseous oxidized mercury (GOM) oxidation of 2.7 months using the GEOS-CHEM model coupled to an ocean general circulation model (MITgcm).

We have modified the suggested sentence slightly as "….(GOM) oxidation" is out of place

"Applying the latest kinetic data, Horowitch et al. (2017) found a lifetime in the atmosphere of GEM with respect to removal by oxidation of 2.7 months using the GEOS-CHEM model coupled to an ocean general circulation model (MITgcm)."

4. Please revise entire manuscript to improve the language.

We have improved the language and two native English-speaking persons have checked the language and made additional improvements.

Dear Editor

285 Dear Editor

Thank you for these last comments. Under each question from editor, I have added the answer in red.

Editor Decision: Publish subject to minor revisions (review by editor) (01 Sep 2020) by Ashu Dastoor

Comments to the Author:

Please revise the manuscript according to my comments below. Thank you

290

Non-public comments to the Author:

1. Please remove "The" at the beginning and add a comma after Station Nord in the title of paper:

Variability in Gaseous Elemental Mercury at Villum Research Station, Station Nord, in North Greenland from 1999 to 2017

295 This has been done in the text.

2. Please correct first sentence, second paragraph, in the introduction to:

The sources of mercury in the environment can be divided into terrestrial emissions (including geogenic, biomass burning and reemissions from soils and vegetation), anthropogenic and oceanic evasion, accounting for roughly 2.1, 2.5 and 3.4 ktonnes of

300 the emissions, respectively (Outridge et al. 2018).

We have changed the text but anthropogenic evasion is not the right term, so instead is add;

"The sources of mercury in the environment can be divided into natural, terrestrial emissions (including geogenic, biomass burning and reemissions from soils and vegetation), anthropogenic, and oceanic emissions, accounting for 2.1, 2.5 and 3.4 ktonnes of the emissions, respectively (Outridge et al. 2018)."

305 3. Please correct the last statement, page 1, in introduction to:

Applying the latest kinetic data, Horowitz et al. (2017) found lifetime of GEM in the atmosphere with respect to gaseous oxidized mercury (GOM) oxidation of 2.7 months using the GEOS-CHEM model coupled to an ocean general circulation model (MITgcm).

310 We have modified the suggested sentence slightly as "….(GOM) oxidation" is out of place

"Applying the latest kinetic data, Horowitch et al. (2017) found a lifetime in the atmosphere of GEM with respect to removal by oxidation of 2.7 months using the GEOS-CHEM model coupled to an ocean general circulation model (MITgcm)."

4. Please revise entire manuscript to improve the language.

We have improved the language and two native English-speaking persons have checked the language and made additional

315 improvements.

[revised manuscript text omitted]
 a negative trend, however not significant atany significant trend at a 90% confidence level.level95% confidence interval (Table 1). The autumn (September-October-November) and the winter months show both negative trends that are significant at a 90% confidence level, (annual and winter data are shown in Figure 5). The trends, in percentage of the average GEM concentrations during these periods, are -1.7%/yr for the winter period and -1.4%/yr for the autumn. The annual trend remains non-significant also when excluding the years 1999 and 2000 or the extreme value in 2017. The lack of a significant annual trend seems to be explained by the high variability of the concentrations during the spring period as well as the fact that the GEM concentration during the summer period show no evidence of a decreasing trend.shows no evidence of a decreasing trend. This lack of yearly trend is the result of a combination of rather different seasonal trends: in the autumn (September October November) there is an insignificant decrease (-0.87%/yr), whereas in winter (December January February) a pronounced decrease of -1.56%/yr is observed (significant on 95% confident interval). A test of the importance of the value in 2000 showed that the decrease is almost unchanged removing the point but R² falls to 0.29 though the trend is still significant. In the spring (March April May) there is not any significant trend, though a small positive trend is seen (0.35%/yr). For the summer (June July August) there is a positive trend of 0.75%/yr that however is not significant. This result is similar to the result obtained at Zeppelin Station on Svalbard for the period 2000 to 2008 (Berg et al. 2013) and, as previously mentioned, at Alert, Canada, where a negative trend of -0.009 ng/m$^3$ (-0.58%/yr) is seen for the period between 1995 and 2008 (Steffen et al. 2015). AIn a study of GEM in firn snow from the Greenlandic inland ice at about 3 km altitude, DommergueDommergueDommercue et al. (2016).) showed that there is a positive trend or no trend during the period 2000–2010, though the authors pointed out that nothing can conclusively can be said about the concentration trends based on their results. The behaviour of the trends may in principle be explained by changes in the emissions in the source regions, in transport patterns, in deposition, re-emission as well as atmospheric chemistry. The seasonal differences in the trends must be explained by a different influence of these factors during the different seasons. Finally, it has been suggested that decreasing GEM concentrations in the Northern Hemisphere over the last 20 years may be partially explained by increased uptake by vegetation due to increased net primary productivity (Jiskra, 2018). Our data set does not permit to evaluate this hypothesis. In the following section, we will discuss these possible explanations for the observed trends separately.

**Commented [HS5]:** To be updated.

**3.1 Changes in atmospheric chemistry**

545 The strongest concentration trend is found during the winter, where photochemically driven chemistry obviously does not take place in the area but where long-range transport from mid-latitudes is at its maximum. The main influence of Arctic atmospheric chemistry on GEM concentrations is expected to be in the spring and summer period, where the fate of GEM is believed to depend on the presence of seasonal sea ice and the presence of air temperatures below -4$^o$ C (Christensen et al. 2004). Figure 6̶5̶ shows a conceptual description of mercury removal in Arctic. A̶l̶s̶o̶ ̶a̶A regression analysis of the number of

550 hours with depletion events (defined here as GEM< 0.5 ng/m$^3$) did not show any significant change over the years 2000–2017. N̶e̶i̶t̶h̶e̶r̶ ̶t̶The̶t̶h̶e̶ ozone data obtained during the period 1999–2017 d̶i̶d̶ ̶s̶h̶o̶w̶ ̶a̶n̶y̶ also showed no significant trend for the concentrations in spring or summer. The ozone observations will be the subject of a separate publication.

The data until 2002 were used to investigate reaction kinetics of ozone and GEM with a third reactant. Log—log plots of ozone against GEM gave a straight line as seen earlier (Schroeder et al. 1998; Berg et al. 2003; Steffen et al. 2008; Skov et al. 2004).

555 A reaction rate for Br with Hg$^0$ was calculated, which fitted well with a reaction rate determined by theoretical chemistry (Goodsite, Plane, and Skov 2012b; Goodsite, Plane, and Skov 2004; Skov et al. 2004). We made the same analysis on the data from 2007 and onwards. GEM was averaged to a time resolution of 0.5 hours.h̶o̶u̶r̶.̶ The new analysis confirmed the previous result, though the data points were more scattered and thus the resulting slope had a higher̶w̶a̶s̶ ̶c̶o̶n̶n̶e̶c̶t̶e̶d̶ ̶w̶i̶t̶h̶ ̶a̶ ̶l̶a̶r̶g̶e̶r̶ 
[revised manuscript text omitted]